

# Comparison of three clustering approaches for detecting novel environmental microbial diversity

Dominik Forster, Micah Dunthorn, Thorsten Stoeck and Frédéric Mahé

Department of Ecology, Technische Universität Kaiserslautern, Kaiserslautern, Germany

## ABSTRACT

Discovery of novel diversity in high-throughput sequencing studies is an important aspect in environmental microbial ecology. To evaluate the effects that amplicon clustering methods have on the discovery of novel diversity, we clustered an environmental marine high-throughput sequencing dataset of protist amplicons together with reference sequences from the taxonomically curated Protist Ribosomal Reference ($PR^2$) database using three *de novo* approaches: sequence similarity networks, USEARCH, and Swarm. The potentially novel diversity uncovered by each clustering approach differed drastically in the number of operational taxonomic units (OTUs) and in the number of environmental amplicons in these novel diversity OTUs. Global pairwise alignment comparisons revealed that numerous amplicons classified as potentially novel by USEARCH and Swarm were more than 97% similar to references of $PR^2$. Using shortest path analyses on sequence similarity network OTUs and Swarm OTUs we found additional novel diversity within OTUs that would have gone unnoticed without further exploiting their underlying network topologies. These results demonstrate that graph theory provides powerful tools for microbial ecology and the analysis of environmental high-throughput sequencing datasets. Furthermore, sequence similarity networks were most accurate in delineating novel diversity from previously discovered diversity.

# INTRODUCTION

High-throughput sequencing technologies have fundamentally changed our perceptions and concepts of environmental protist diversity (*Amaral-Zettler et al., 2009*; *De Vargas et al., 2015*; *Logares et al., 2014*; *Massana et al., 2015*; *Stoeck et al., 2009*). Current high-throughput sequencing surveys analyze protist communities by targeting specific molecular markers, resulting in datasets of many millions of sequencing reads that can be used to address community-comparative, ecosystem-functioning, and novel-diversity questions (*Dunthorn et al., 2014b*). The detection of novel diversity, in specific, is often based on sequence similarity. Potentially novel reads are identified by having a low similarity to previously sequenced reference taxa (e.g., *Berney et al., 2013*; *Dunthorn et al., 2014b*; *Edgcomb et al., 2011b*; *Filker et al., 2014*; *Gimmler & Stoeck, 2015*; *Hartikainen et al., 2014*). Following this strategy, groups of sequences that contain both environmental reads and

Corresponding authors
Dominik Forster, dforster@rhrk.uni-kl.de
Frédéric Mahé, mahe@rhrk.uni-kl.de

references represent environmental diversity which is covered by taxonomic reference databases, whereas groups of sequences that exclusively contain environmental reads represent novel variants of diversity. Our understanding of protist diversity is far from complete (*Pawlowski et al., 2012*). While the detection and description of novel protists is a central task, our ability to detect novel diversity in molecular environmental studies is affected by the way reads are clustered into operational taxonomic units (OTUs).

A traditional method of constructing *de novo* OTUs is by using the popular program USEARCH (*Edgar, 2010*), though several other similar alternatives exist (e.g., *Fu et al., 2012*; *Ghodsi, Liu & Pop, 2011*; *Schloss et al., 2009*). USEARCH and these other related programs initiate OTUs by selecting an amplicon (i.e., a dereplicated read) to serve as a centroid. Pairwise comparisons score the global sequence similarity of other amplicons with the centroid. Amplicons with a global sequence similarity to the centroid equal or greater than a given threshold join the OTU. The OTU is then closed, and its maximal radius (or diameter, depending on the method used) is equal to the global similarity threshold value. There is no consensus on which global similarity threshold value should be used because taxa evolve at different rates (*Brown et al., 2015*; *Caron et al., 2009*; *Nebel et al., 2011*): a 97% value is commonly used in protist studies (*Edgcomb et al., 2011a*; *Massana et al., 2015*), although higher values are also used (*Egge et al., 2015*).

A second method of constructing *de novo* OTUs is by using sequence similarity networks (e.g., *Forster et al., 2015*). Each node in these networks represents one amplicon, and two nodes are connected by an edge only if their amplicons are within a global similarity value that is computed by pairwise alignment scores. Sequence similarity networks seldom result in one single continuous graph, but consist of several subgraphs of connected nodes. These subgraphs are called connected components and can be used as OTUs (*Forster et al., 2015*). Since additional nodes are added iteratively, the radius of a connected component is not pre-defined, but can be any value, including higher than the global similarity value. As with USEARCH, there is no agreement upon which global similarity threshold should be used. Unlike USEARCH, sequence similarity networks produce OTUs that exhibit an internal network topology which can be further evaluated by methods of graph theory (*Bapteste et al., 2012*; *Bittner et al., 2010*; *Jachiet et al., 2013*; *Junker & Schreiber, 2011*; *Newman, 2010*). For instance, assortativity analyses reveal if nodes that share the same trait preferentially connect with each other (*Newman, 2003*), while centrality analyses give information about the position of a node in a network and if this node serves as a hub in the network (*Newman, 2005*). How these and other methods can address ecological questions in high-throughput sequencing diversity surveys of protists is demonstrated in *Forster et al. (2015)*. To target novel diversity, we used shortest path analyses as a straightforward way to measure the distance (expressed as the number of edges that have to be crossed) between two nodes in a network (*Alvarez-Ponce et al., 2013*; *Forster et al., 2015*), specifically between environmental amplicons and reference sequences.

A third method to define *de novo* OTUs is by using the program Swarm (*Mahé et al., 2015*; *Mahé et al., 2014*). Unlike USEARCH, Swarm relies on an iterative, single-linkage algorithm that uses a small local clustering threshold $d$. This value $d$ is user-defined (1 by

default), and corresponds to the maximum number of differences due to substitutions or insertions/deletions between two globally aligned amplicons. Swarm selects the most abundant amplicon available in the amplicon pool to serve as a centroid for a new OTU. All pool amplicons with $d$ or less differences to the centroid are added to the OTU and removed from the pool. Each of those newly added amplicons are compared to the pool amplicons to find those with $d$ or less differences. The process is repeated until no new amplicon can be added to the OTU. To avoid the formation of long chains of amplicons, a classic issue with single linkage clustering, swarm takes into account the abundance of each amplicon (i.e., the number of times it has been observed) and can interrupt the iterative process locally. The combination of these two processes confers to Swarm a high level of stringency and a robustness to changes in initial conditions (e.g., order and abundance of amplicons). Like sequence similarity networks, Swarm produces OTUs whose radii can be any value. Also like sequence similarity networks, the internal connections between the amplicons in Swarm's OTUs create a network of edges and nodes, which can be evaluated using methods based on graph theory.

To compare how USEARCH, sequence similarity networks, and Swarm affect our ability to uncover novel diversity in protists, we used amplicon data derived from samples taken in European coastal marine environments. To place the environmental amplicons into a taxonomic context of already known diversity, we relied on the curated Protist Ribosomal Reference database (PR$^2$) (*Guillou et al., 2012*). With this combination of environmental and taxonomically-identified amplicons, we asked: (i) Do all three clustering approaches predict the same amount of novel diversity? (ii) Do network analyses uncover additional novel diversity within OTUs that have underlying network topologies?

## MATERIAL AND METHODS

### Datasets

We used already published environmental high-throughput sequencing data from the BioMarKs Consortium (www.biomarks.eu) that sampled microbial eukaryote communities at six near-shore marine sites in Norway, France, Spain, Italy and Bulgaria (e.g., *Bittner et al., 2013*; *Dunthorn et al., 2014a*; *Logares et al., 2014*; *Massana et al., 2015*). The sample design and sample processing, as well as Roche/454 GS FLX Titanium sequencing of the V4 region of 18S rDNA, is detailed in *Massana et al. (2015)*. Quality filtering and chimera check of the raw reads with both UCHIME (*Edgar, 2010*) and ChimeraSlayer (*Haas et al., 2011*) is also outlined in *Massana et al. (2015)*. The 1,476,249 cleaned V4 DNA and RNA reads were dereplicated into 312,503 strictly identical amplicons (dataset provided as Supplemental Information 1). The scripts used to perform the analyses presented in this study can be found online in HTML format (File S1).

For reference amplicons, we used the PR$^2$ v203 taxonomic reference database (*Guillou et al., 2012*). From this database we extracted 115,043 taxonomically identified V4 amplicons. We then combined these reference amplicons with the environmental amplicons for all downstream analyses. The clustering approaches would thus produce OTUs containing

a blend of environmental and reference amplicons, or OTUs containing only one type of amplicons. Our goal was to identify differences between the three approaches, focusing on OTUs containing exclusively environmental amplicons, more likely to represent novel diversity.

## Clustering

Three *de novo* clustering approaches were used to cluster the combined amplicons. First, USEARCH v8.0.1623 (*Edgar, 2010*), with a 97% global similarity value using options *-cluster smallmem* and *-sortedby size*. This analysis took 57 s on a Linux 2.6 operated machine with dual Intel Xeon E5-2670 processors (2.6 GHz) using 16 physical cores and 64 GB RAM.

Second, basic network topology information was gathered by running a global pairwise alignment analysis in VSEARCH v1.1.3 (https://github.com/torognes/vsearch) using options *-allpairs_global* and *-iddef 1*. This analysis took about 5 days on the same computer. The resulting matrix contained 682,621,198 edges with a weight of at least 97% global sequence similarity. Based on this matrix we created sequence similarity networks in R version 3.2.1 (http://r-project.org) using 'igraph' scripts (*Csardi & Nepusz, 2006*). To allow these network analyses in 'igraph,' we had to switch to a Linux 2.6 operated machine with dual Intel Xeon E5-4650 processors (2.7 GHz) using 32 physical cores and 256 GB RAM. Usage of less memory or more input data forced an untimely abort of analyses. The sequence similarity networks analyses took 3 h with this setting.

Third, SWARM v2.1.1 (*Mahé et al., 2015*; *Mahé et al., 2014*), with $-d = 1$ and $-f$. This analysis took 33 s on the first computer. Singleton and doubleton OTUs (OTUs consisting of one or two amplicons, respectively) were removed from the results of all three clustering approaches for downstream analyses.

## Analyses

For each clustering approach we distinguished if an OTU consisted of: (i) both environmental and reference amplicons, (ii) exclusively reference amplicons, and (iii) exclusively environmental amplicons. The number of reads in each OTU was also counted.

To compare the novel diversity reported by each clustering approach, we analyzed OTUs consisting of exclusively environmental amplicons. For each amplicon in exclusively environmental OTUs, we conducted global pairwise alignments of these amplicons with all PR$^2$ references using VSEARCH (with the options *-allpairs_global*, *-iddef 1* and *-id 0.70*), and recorded the highest percentage of similarity to any reference. This revealed how divergent the novel diversity reported by each clustering approach was with regard to taxonomically identified references. Considering that a 97% sequence similarity threshold is routinely used to delineate between different protist species (*Edgcomb et al., 2011a*; *Massana et al., 2015*), we applied the same threshold and expected each potentially novel diversity amplicon in exclusively environmental OTUs to be less than 97% similar to its closest reference. We also compared if the same environmental amplicons were classified as novel diversity among the different approaches.

To analyze the internal network structure of OTUs, shortest path analyses were conducted within each sequence similarity networks and within each Swarm OTU with

**Table 1 Sequence clustering results of the three tested approaches.** Indicated is the amount of OTUs and the amount (and type) of amplicons within these OTUs for each class of OTUs defined in our analyses.

| | USEARCH | Sequence similarity networks | Swarm |
|---|---|---|---|
| OTUs | 12,427 | 8,202 | 13,240 |
| OTUs containing environmental and reference amplicons | 2,527 | 1,619 | 1,993 |
| *Environmental amplicons* | 223,735 | 253,965 | 142,946 |
| *Reference amplicons* | 33,386 | 54,988 | 18,774 |
| OTUs containing exclusively reference amplicons | 4,558 | 3,138 | 5,019 |
| *Reference amplicons* | 59,368 | 46,255 | 49,147 |
| OTUs containing exclusively environmental amplicons | 5,342 | 3,445 | 6,228 |
| *Environmental amplicons* | 71,337 | 47,116 | 81,073 |

'igraph' scripts. The shortest path concept emerges from graph theory and exploits connections between nodes in a network (*Newman, 2010*). In this particular case, we used shortest path analyses to find the minimal number of edges (i.e., connections) that have to be crossed within an OTU to move from each environmental node (i.e., amplicon) to its closest reference node. If an environmental node and a reference node were directly linked (i.e., direct neighbors separated by exactly one edge), they exhibited a distance of '1' to each other. As the edges reflected global sequence similarity values of at least 97% (in sequences similarity networks), or a local basepair difference of '1' (in Swarm), we defined these environmental nodes as the part of diversity that is well represented by the $PR^2$ reference database. Environmental nodes that were not directly linked to reference nodes exhibited a distance of two edges or more, and were thus indirectly linked. Environmental nodes in OTUs, which exclusively consisted of environmental amplicons, exhibited 'infinite' distances to all reference nodes since no shortest path existed. We defined all environmental nodes with distances of two edges or more to reference nodes as novel variants of diversity that are currently not covered by the $PR^2$ database.

## RESULTS AND DISCUSSION

### Contrasting OTU results from three approaches

The number of resulting OTUs varied across the three clustering approaches (Table 1). The fewest OTUs in total were produced by sequence similarity networks ($n = 8,202$). Sequence similarity networks also produced the fewest OTUs containing both environmental and reference amplicons ($n = 1,619$), containing exclusively reference amplicons ($n = 3,138$), and containing exclusively environmental amplicons ($n = 3,445$). This approach was especially effective in linking environmental and reference amplicons: it had the most amplicons in OTUs containing both types ($n = 253,965$ environmental and $n = 54,988$ reference). On the other hand, this also led to fewer amplicons in exclusively environmental OTUs ($n = 47,116$), meaning that sequence similarity networks reported the least novel diversity in terms of both amplicons and OTUs.

USEARCH produced more OTUs in total ($n = 12,427$) and more OTUs ($n = 5,342$) that contained exclusively environmental amplicons ($n = 71,337$). The fraction of novel amplicons was therefore increased by one third in USEARCH compared to sequence similarity networks. These differences in OTU numbers may be due in part to how the two methods use their global clustering values: while connected components in sequence similarity networks grow iteratively, OTUs in USEARCH are restricted to a maximum radius (at most 3% divergence from the centroid). Amplicons whose sequences are less than 97% similar to the centroid are consequently placed outside of the OTU, although they might be more than 97% similar to other amplicons inside the OTU. This behavior of USEARCH and other closely-related methods results in an over-splitting of OTUs (*Flynn et al., 2015*; *Mahé et al., 2014*) compared to sequence similarity networks. Additionally, this behavior also causes OTU instability, meaning that a re-clustering with USEARCH may result in slightly different OTU sizes and membership, especially if the input order of the amplicons is shuffled (*He et al., 2015*; *Mahé et al., 2014*). Since both factors are especially important for an accurate detection of novel diversity, we argue that the more conservative results of the sequence similarity networks are less prone to contain amplicons and OTUs that are spuriously classified as novel.

Although not tested here, previous studies have shown that all-vs.-all pairwise comparison clustering approaches such as sequence similarity networks generally produce more reliable and stable OTUs than heuristic clustering methods such as USEARCH (*Schmidt, Matias Rodrigues & Von Mering, 2015*; *Sun et al., 2011*). This higher reliability and stability of all-vs.-all pairwise comparisons comes at the cost of extensive computational time (*Flynn et al., 2015*; *Sun et al., 2011*), which increases with the square to the number of input sequences (*Bik et al., 2012*). By calculating a pairwise comparison matrix of the currently largest dataset of near-shore marine protists in Europe, we operated close to the limit of dataset size that can be handled in all-vs.-all current approaches.

Compared to both approaches relying on global clustering values, Swarm, with its local clustering value, produced the most OTUs in total ($n = 13,240$). The Swarm approach also produced the most OTUs ($n = 6,228$) that contained exclusively environmental amplicons ($n = 81,073$). These higher numbers of OTUs in total and OTUs containing exclusively environmental amplicons may be due to Swarm's high clustering stringency that iteratively links amplicons with a small number of differences to each other. On the other hand, these high numbers may be due to missing intraspecific sequence variation in the PR$^2$ reference database, which usually contains only one reference per species. In natural communities, intraspecific genetic variation of microbial organisms may be much more diverse than just a few base pair differences, especially in hypervariable gene regions (*Brown et al., 2015*; *Decelle et al., 2014*; *Dunthorn et al., 2012*; *Pernice et al., 2013*). But in Swarm, an environmental amplicon that differs by more than one base pair to a reference sequence will be placed into a novel OTU, if there are no intermediate amplicons linking them. As long as reference databases are not covering intraspecific sequence variation, it is a more effective strategy to compute Swarm OTUs from datasets consisting entirely of environmental amplicons, and perform a later taxonomical assignment; e.g., as in *De Vargas et al. (2015)*, *Filker et al. (2014)* and *Gimmler & Stoeck (2015)*.

## Is novel diversity really novel?

After the identification of novel variants of OTUs and amplicons, the next step in the discovery of novel diversity is normally the design of specific primers and probes for the targeted recovery of organisms from environmental samples (*Edgcomb et al., 2011b*; *Gimmler & Stoeck, 2015*; *Hartikainen et al., 2014*; *Orsi et al., 2012*; *Seenivasan et al., 2013*). However, this process is time-, cost-, and labor-intensive. An accurate initial classification of novel diversity by clustering approaches is therefore crucial.

There were 29,059 environmental amplicons that were classified as novel by all three clustering approaches (Fig. 1). However, the number of environmental amplicons classified as novel exclusively by one approach differed dramatically: 1,232 in sequence similarity networks, 13,777 in USEARCH, and 40,132 in Swarm. Most environmental amplicons which shared less than 97% sequence similarity with references in $PR^2$ were congruently classified as novel by all three approaches. But both USEARCH and Swarm classified as novel numerous amplicons that were more than 97% similar to $PR^2$ references (Fig. 2, Fig. S1). Even though clustering in USEARCH was performed at 97% similarity to delineate novel environmental amplicons from amplicons representing previously described diversity, we found 15,438 amplicons in exclusively environmental OTUs with more than 97% similarity to $PR^2$ references; for Swarm this fraction amounted to 47,007 amplicons. The even larger estimation of novel diversity by Swarm is caused by a combination of the approach's high clustering stringency and missing intraspecific variation in the $PR^2$ database. On the other hand, sequence similarity networks classified no environmental amplicon inadvertently as novel, thereby supporting our argument of more accurate novel diversity detection in the latter approach. We conclude that the conservative results of sequence similarity networks most closely match our definition of how we delineated novel diversity from previously described diversity, for a given global clustering threshold value.

Beyond that, 97% of the novel diversity amplicons in sequence similarity networks were identified as novel by at least one of the other two clustering approaches (Fig. 1). On the other hand, the 1,232 amplicons exclusively identified as novel by sequence similarity networks clustered into singletons or doubletons in USEARCH and Swarm and were thus excluded from downstream analyses. The novel diversity uncovered by sequence similarity networks therefore comes closest to a subset of amplicons detected by all three clustering approaches that is truly less than 97% similar to references in $PR^2$. Furthermore, we strongly advise to perform an additionally taxonomic assignment step in Swarm and USEARCH to validate if potential novel diversity is indeed highly diverse from deposited references. At the same time, though, we are aware that even amplicons which are highly similar to entries in reference databases may represent novel genetic variants. Such hidden diversity is unlikely to be unveiled by approaches solely relying on global similarity values. Instead, more stringent approaches that trace local substitutions or methods which explore internal OTU structure stand a higher chance of revealing novel genetic variants, since they provide a higher resolution of genetic diversity.
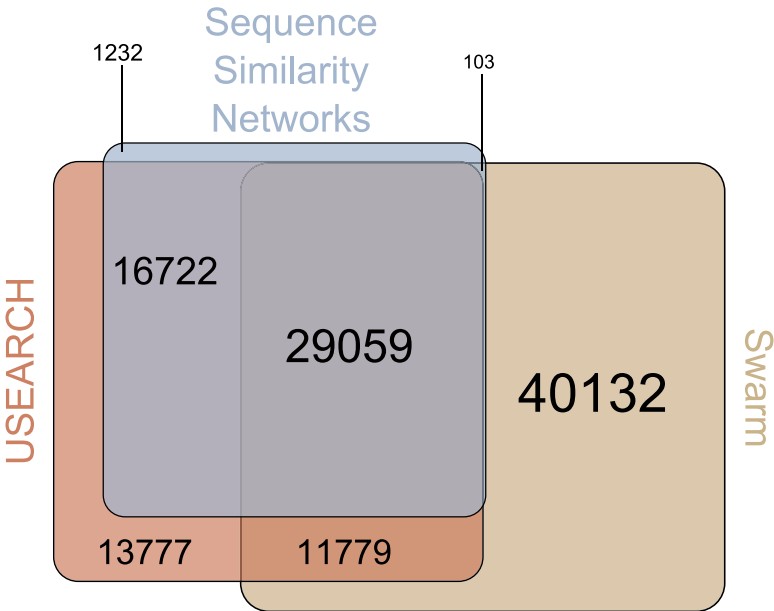

**Figure 1** **Venn-Diagram of the number of amplicons in exclusively environmental OTUs.** The area of each clustering approach was proportionally adjusted to the amount of amplicons in exclusively environmental OTUs detected in that approach. Overlapping areas reflect amplicons detected in each of the respective approaches. Numbers indicate how many amplicons are represented by each area, whereas each area's size is proportional to the number of amplicons included.

## Graph theory allows a more detailed evaluation of high-throughput sequencing datasets

Beyond just being able to relay the number of OTUs, sequence similarity networks and Swarm provided additional underlying information for each of their OTUs in the form of network topologies. As pointed out by *Forster et al. (2015)*, these network topologies can reveal additional within-OTU connections among environmental and reference amplicons by using shortest path analyses.

In sequence similarity network OTUs containing both types of amplicons, 239,472 of the 253,965 environmental amplicons were directly connected to reference amplicons (Fig. 3), while the remaining 14,493 environmental amplicons were indirectly connected to reference amplicons. These latter environmental amplicons represent potentially novel genetic variation on top of the 47,116 amplicons placed into sequence similarity network OTUs which contained no reference amplicons. In Swarm OTUs that contained both types of amplicons, only 5,757 of the 142,946 environmental amplicons were directly connected to reference amplicons. The 137,189 environmental amplicons with indirect connections also represent novel genetic variation along side of the 81,073 amplicons in exclusively environmental OTUs. This large number of indirectly connected amplicons in Swarm OTUs may be an overestimation because current reference databases do not yet cover intraspecific sequence variation (see above). However, our analyses are a first indication that shortest path analyses are a promising way to explore Swarm OTUs. By analyzing paths within an OTU one could, for example, investigate whether amplicons

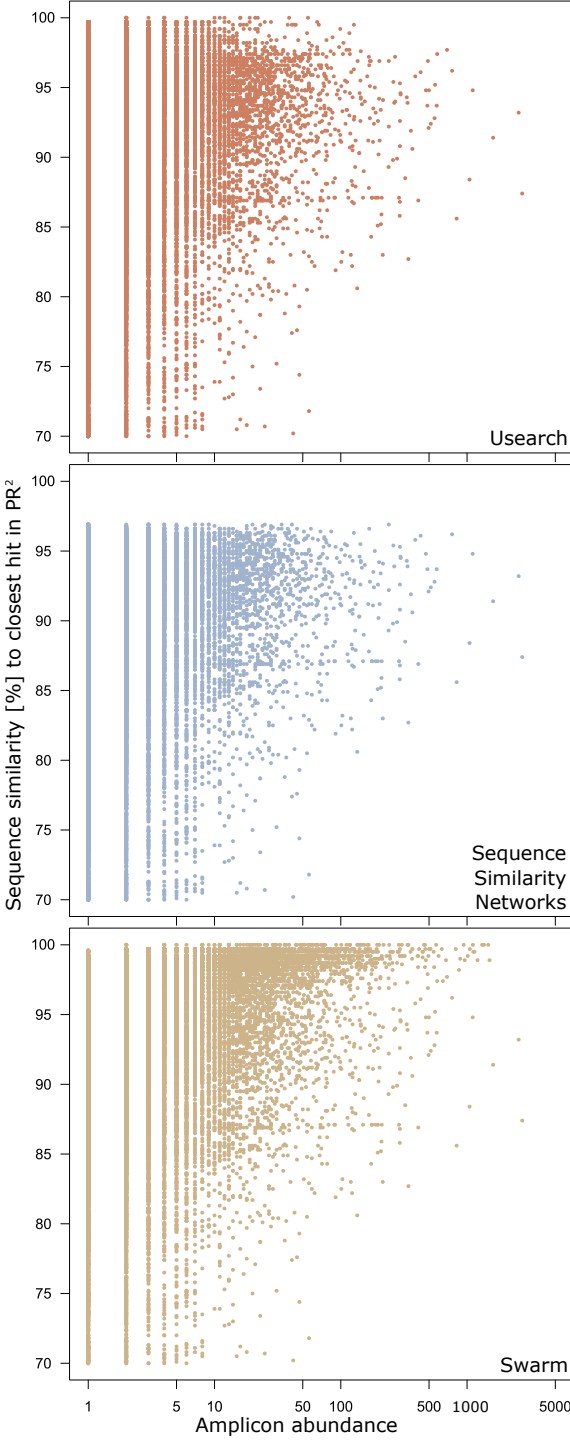

**Figure 2   Genetic divergence of amplicons in exclusively environmental OTUs to PR² references by clustering approach.** Each point represents one amplicon clustered into an exclusively environmental OTU by the respective clustering approach. Position on the $x$-axis gives the abundance of each amplicon in the initial dataset before dereplication. The $y$-axis gives the highest pairwise sequence similarity score of an amplicon to any entry in the PR² database as calculated by VSEARCH.

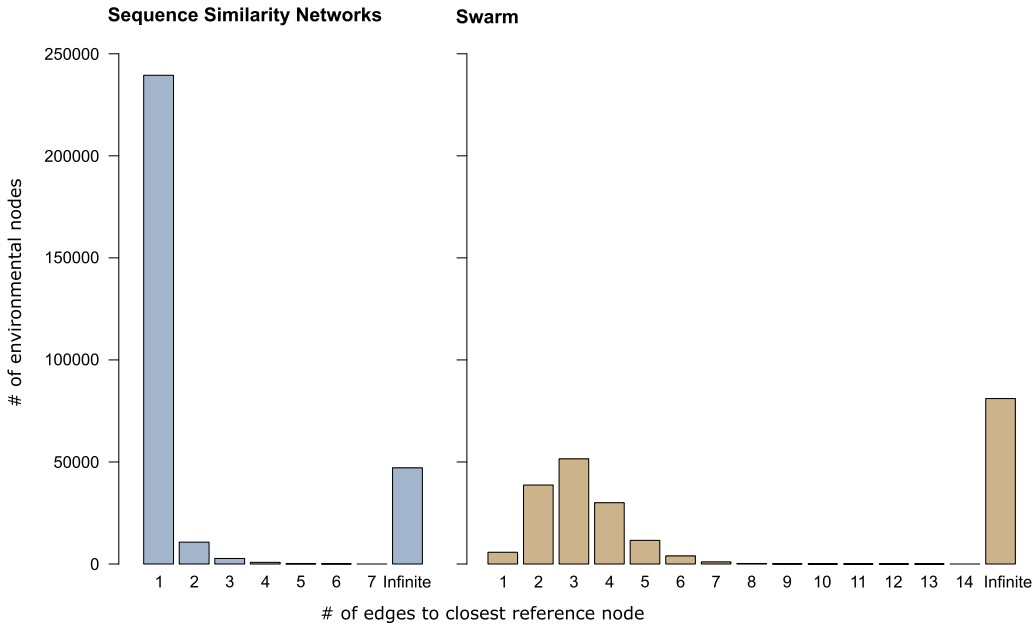

**Figure 3 Shortest path analyses of CCs and swarms.** The plots illustrate how many edges separated each environmental amplicon from its closest reference amplicon in sequence similarity networks and Swarm. A distance of '1' edge means that the environmental amplicon was directly connected to a reference. 'Infinite' means that the environmental amplicon was placed into an exclusively environmental OTU (see also Table 1) and did not exhibit any connection to a reference amplicon.

from the same sampling site are more often directly connected to each other than to amplicons from another site. Thus, screening for genetic variation related to regional populations or species.

Nevertheless, shortest path analyses are just one way to explore genetic variance and novel diversity within OTUs with network topologies. Graph theory can be used to ask numerous questions in microbial ecology (*Junker & Schreiber, 2011*; *Newman, 2010*; *Proulx, Promislow & Phillips, 2005*). For instance, analyses of assortativity can indicate if environmental sequences affiliated with a certain habitat more preferentially connect with reference sequences than environmental sequences affiliated with another habitat (*Forster et al., 2015*), thus revealing which habitat's microbial community is less adequately covered by reference databases.

## CONCLUSIONS

Each of the three clustering approaches provided different perspectives on microbial diversity, while also showing individual weaknesses. Our results corroborate previous observations of inaccuracy in heuristic clustering approaches and highlight how this inaccuracy also affects the detection of novel diversity. Despite their weaknesses, we argue that the combination of high stringency clustering methods and sequence similarity networks, and the implementation of additional tools based on graph theory principles will be beneficial for the evaluation of high-throughput sequencing datasets. Such tools

will uncover underlying patterns from microbial high-throughput sequencing data, which hold important information about environmental microbial communities.

## ACKNOWLEDGEMENTS

We would like to thank the computational resources at the Regional Computing Center at the University of Kaiserslautern, and the BioMarKs consortium for the data analyzed in this study. We thank the editor, Antonio Fernandez-Guerra, and an anonymous reviewer for constructive comments.

### Funding

FM and MD were supported by the Deutsche Forschungsgemeinschaft (grant #DU1319/1-1). DF was supported by a graduate scholarship of Stipendienstiftung Rheinland-Pfalz. TS was supported by the Deutsche Forschungsgemeinschaft (grant #STO/414/11-1). The funders had no role in study design, data collection and analysis, decision to publish, or preparation of the manuscript.

### Grant Disclosures

The following grant information was disclosed by the authors:
Deutsche Forschungsgemeinschaft: #DU1319/1-1, #STO414/11-1.
Stipendienstiftung Rheinland-Pfalz.

### Competing Interests

The authors declare there are no competing interests.

### Author Contributions

- Dominik Forster conceived and designed the experiments, performed the experiments, analyzed the data, wrote the paper, prepared figures and/or tables, reviewed drafts of the paper.
- Micah Dunthorn and Thorsten Stoeck conceived and designed the experiments, contributed reagents/materials/analysis tools, wrote the paper, reviewed drafts of the paper.
- Frédéric Mahé conceived and designed the experiments, performed the experiments, contributed reagents/materials/analysis tools, wrote the paper, reviewed drafts of the paper.

### Data Availability

The research in this article did not generate any raw data.

### Supplemental Information

Supplemental information for this article can be found online at http://dx.doi.org/10.7717/peerj.1692#supplemental-information.

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
