# Peer review of "Comparison of three clustering approaches for detecting novel environmental microbial diversity"

_PeerJ, doi:10.7717/peerj.1692_

## Round 0.1 · original submission · Major Revisions

As you will see, both reviewers have a number of helpful comments, and some of these are rather major comments regarding both style and content. Please consider the comments carefully. Considering the nature and extent of the comments by both reviewers, I cannot guarantee that the manuscript will be accepted after revision.

·

Basic reporting

No comments. Looks good.

Experimental design

No comments. Looks good.

Validity of the findings

The authors kindly provide the code used for the different analyses.
For reproducibility, the authors should provide the data after the pre-procesing steps and specially the sequence similarity network weighted edge list for the igraph network analyses.

Additional comments

In this manuscript, Foster et al. do a comparison of three different cluster approaches to analyse the microbial diversity of environmental samples and how the different methods can modify our vision in terms of amplicon novelty in the environment. In my opinion, a very interesting and important topic in microbial diversity studies, as defining de novo OTUs is a crucial step that could influence all downstream analyses and consequently affect the interpretation of the results obtained.

They show how sequence similarity networks results in more stable and reliable OTUs, this approach that few years seemed prohibitive in terms of computational requirements, nowadays with the algorithmic improvements like the ones implemented in VSEARCH make them feasible. As the authors stated in the manuscript,

"By calculating a pairwise comparison matrix of the currently largest dataset of near- shore marine protists in Europe, we operated close to the limit of data that can be handled in all-vs.-all current approaches.”

it would be interesting that the authors add the computational times and the hardware used by the three methodologies so the reader would have an idea of what implies the analysis of such large datasets.

One interesting result from the manuscript is how sequence similarity networks reduce the over-splitting of OTUs observed in widespread methods like USEARCH. OTU picking is a crucial step in many diversity studies, would be highly interesting to investigate how the different clustering methods compared in this work affects the different diversity indices and to the overall community structure of the samples.

Another important issue the authors are tackling is the identification of novel amplicons, not surprisingly sequence similarity networks reported the less amount of novel amplicons due the implicit continuos nature of the network. The authors would emphasize that sequence similarity networks are not only a good proxy to study the diversity but that they are highly interesting to track the evolutionary paths of the different OTUs members and can be used as an additional source of information to get a better understanding of the observed diversity.

One thing I miss in the manuscript is a more detailed network analysis by the authors in a similar fashion as they already did in Foerster et al. 2015. I think that presenting briefly the results of the assortativity and path length analyses makes no justice to the power of the network analysis approach applied to sequence similarity networks or Swarm OTU clustering. Another good point would be to add centrality and clique analyses to unveil other properties of the networks and the important role that certain amplicons would play among the OTUs. In addition a more detailed description of the network topology would be helpful to have a better understanding of sequence similarity networks or Swarms; as an example, referring to the assortativity analysis used in the manuscript, in overall, can the sequence similarity networks be classified as disassortative or assortative? Do we observe the same for Swarms?

Despite the limitations described by the authors in applying Swarms to the experimental setup described in the manuscript (missing intraspecific sequence variation in the PR2 reference database), in Figure 1, are described the number of amplicons shared between the different approaches, do the authors think if it would be valuable to compare the level of intersection between the Swarm results and sequence similarity networks? In other words, how many connections are common between the shared amplicons in both networks.

Reviewer 2 ·

Basic reporting

* The submission must adhere to all PeerJ policies.
The authors use readily available datasets and have accompanied their submission with code to allow reproducibility.

* The article must be written in English using clear and unambiguous text and must conform to professional standards of courtesy and expression.

The authors should have this manuscript reviewed to improve the quality of the English. The errors are common enough to impede reading:
== Rephrase ==
Abstract: "an environmental marine protist HTS dataset of protist reads"
45: Please rephrase this: "community-comparative, ecosystem-functioning, and novel-diversity questions" to something like "questions of community composition, ecosystem functioning, and microbial diversity". In general, avoid over-using compound modifiers.
46: "in specific" --> "in particular"
49-53: restructure for readability.
66: "stands for" --> represents
71: ... including higher than the global ...
85: similar
88: ... Also like sequence similarity networks --> ... and ...
89: full stop
92-93: rephrase, esp. "already known diversity"
93: "Accessions" is not really correct. The authors obtained sequences from the database. Those sequences *were* accessions when they were submitted to and added to the database (and issued accession numbers).
97: "novelty diversity"
102: already published
---- At this stage, I have stopped making suggestions. The authors should proof-read this manuscript very carefully or have it revised to improve the language.


* The article should include sufficient introduction and background to demonstrate how the work fits into the broader field of knowledge. Relevant prior literature should be appropriately referenced.
This is generally the case, but more background on network analysis is needed to make the rationale for their shortest path analyses more clear. As it stands, it seems arbitrary and without context.

*The structure of the submitted article should conform to one of the templates. Significant departures in structure should be made only if they significantly improve clarity or conform to a discipline-specific custom.
I believe this manuscript would benefit from the separation of the results and discussion.

* Figures should be relevant to the content of the article, of sufficient resolution, and appropriately described and labeled.
In general these are valid and informative, however:
Figure 1: While this is useful, I'd move this to a supplement and have a figure showing the graph structure of OTUs generated by Swarm and SSN as the leading figure. This should then be used to propel a discussion on graph analysis on OTUs.

*The submission should be ‘self-contained,’ should represent an appropriate ‘unit of publication’, and should include all results relevant to the hypothesis. Coherent bodies of work should not be inappropriately subdivided merely to increase publication count.
On this front, this submission makes some very valuable observations and supports an important transition in the methodology used to understand diversity using OTU-based methods. However, it diverts momentum from its main (and concluding) point: the material should be refocused on the analysis of graph objects rather than a general (and far less interesting, given that work has been done on the comparison of USEARCH and other methods to methods like Swarm) comparison.

* All appropriate raw data has been made available in accordance with our Data Sharing policy.
The data is available externally, although the authors could include a script to reproduce the data downloads exactly.

Experimental design

* The submission must describe original primary research within the Scope of the journal.
This methodological paper could be more securely within scope if it embraced the exploration of network analyses on OTUs more fully. Currently, this is only alluded to with a single approach used.

* The submission should clearly define the research question, which must be relevant and meaningful. The knowledge gap being investigated should be identified, and statements should be made as to how the study contributes to filling that gap.
This aspect could be strengthened in this submission: the treatment of "novel OTUs", the rationale behind the application of network analyses, and the

* The investigation must have been conducted rigorously and to a high technical standard.
This is the generally the case, however, it is unclear why some thresholds have been chosen (lines 156-158) and, unless a clear justification for such thresholds can be provided, a range of thresholds should be used to evaluate the techniques.

* Methods should be described with sufficient information to be reproducible by another investigator.
This is generally the case, although the parameters of a few approaches are not reported fully (e.g. VSEARCH). The actual commands used could be included in the supplementary files (and this indicated clearly in the text).

* The research must have been conducted in conformity with the prevailing ethical standards in the field.
There are no issues here.

Validity of the findings

* The data should be robust, statistically sound, and controlled.
The data used is reasonable; however, the results would be much more convincing if simulated data was used alongside real data. It's quite hard to make conclusive statements in a "wild" data set. Also, deeper insight into the graph properties of the OTUs generated by SSN and Swarm would be highly desirable.

* The data on which the conclusions are based must be provided or made available in an acceptable discipline-specific repository.
This is the case. As said before, the authors could provide a script to download the specific data sets they used.

* The conclusions should be appropriately stated, should be connected to the original question investigated, and should be limited to those supported by the results.
This is where the manuscript suffers somewhat - the conclusions speak to the applicability of graph analyses to the evaluation of "novel" or unclassified diversity in OTUs, however, much of the manuscript focuses on a more general comparison of three methods, one of which is not amenable to graph analyses.

Additional comments

== Abstract ==

I would not describe the discovery of "novel diversity" as a central task of environmental microbial ecology. We should be able to accurately detect whatever diversity is present, novel or not. I believe the latter is what swarm excels at.

"with accessions" - please be more precise here, we're still talking about sequences correct?

Phrases like "novel diversity OTUs" are confusing: what exactly is meant here?

"Using graph theory": graph theory is a rather large discipline, please be more specific about what you applied. Also, graph theory is not a "tool". In the closing sentence, the authors seem to suggest that the "graph theory" approach they use does not need further validation. This seems unlikely.


== Introduction ==


47-49: Given the number of references cited and the centrality of the task to the manuscript, I feel that one or two more lines can be dedicated to describing the main principles behind the detection of novel diversity.


54-55: A program is not a method. The authors describe the method a few lines later.

55-56: what makes these alternatives similar? Perhaps best to mention this after describing the method.

57: An amplicon is not a dereplicated sequence, as the parenthetical statement suggests.

58: define what is meant by centroid here.

58: The idea of "distance" between amplicons should be explained more clearly and connected to the idea of pairwise similarity score. It's used here and in the description of the other methods but is not described well.

58-61: this should be rephrased for clarity and expanded slightly, e.g. "Pairwise comparisons score the similarity between the centroid and other amplicons. Amplicons with scores above a certain threshold are added to the OTU ... "

69 (and in general): there is an overuse of the modal verb, "can". "Groups of nodes which form enclosed connected components are considered OTUs" is more understandable. Also, "connected component" should be explained for the uninitiated.

74-75: I don't think the authors mean "graph theory analyses", but rather "analytical approaches native to graph theory" or similar.

77-83: these lines could be restructured to give a more clear idea of what Swarm does. Start with the end of line 80 "Swarm begins by..." and then define d and describe how the single-linkage algorithm proceeds.

85-86: what is meant by "attract" here? how can an amplicon attract another amplicon? In general, the authors could benefit from clarifying their (useful) abstractions.

86-87: the robustness of Swarm is quite an important point and should have at least one sentence dedicated to it, rather than appear as a dependent clause.

88-89: here "links" is used, rather than "edges" as introduced in the discussion of SSNs. This may be confusing and would require some more description.

92: Organisms have habitats, not physiographic features - "European coastal marine environments" is correct.

92-94: obtaining sequences from a database does not explain how the authors placed environmental sequences into a "taxonomic context". Please expand on this.

95: "unknown-environmental" makes little sense. The authors mean environmentaly amplicons with no taxonomic affiliation.

97-98: graph theory cannot discover anything - some of the methods based on graph theory can be used to detect "novel diversity". Even at this stage, it is not clear what the authors mean by novel diversity - this should be very clearly stated early on in the introduction. Further, no OTUs have underlying network topologies unless some method asserts one. Do the authors mean to transition to network theory as opposed to graph theory to discuss topologies? The former is a part of the latter with specific constraints.


== Materials and methods ==

102-105: "For environmental ... 2015)." I assume the authors mean that these were the environmental amplicons analysed. Rephrase for clarity.

107-109: Any options and parameters used for UCHIME and Chimeraslayer should be described.

114-115: A line describing why the environmental and PR2 sequences were combined would be useful here.

118-126: Some rationale on why those parameters where chosen for this comparison is needed. What makes these combinations of parameers suitable for a fair comparison of these methods? This can be done either here on in the discussion.

126: this line contains a fragment of a sentence.

126-128: singleton and doubleton OTUs should be defined. I know most microbial ecologists who have dealt with sequence data will know about these, but the manuscript should stand alone where possible.

137 and elsewhere: "environmental [OTU, amplicon, node]" sounds wrong - all OTUs are derived from organisms in some environment. Perhaps "unclassified OTU" or similar would be more accurate.

136-140: Why was VSEARCH used and why those parameters?

139-140: "and gathered...reference sequence". This should be explained more clearly as well as what the results are and how they were interpreted. The current treatment is too cursory.

149: It would be useful to remind the reader what a node is in this context.

156-158: Why? What rationale do the authors base this threshold on? Is there any ecological basis for this? Is it an arbitrary threshold used for testing? If the latter is true, then what is the impact of changing this? Does the behaviour of the methods change? If the authors have no ecological basis for using such a threshold, then it would be advisable to include multiple thresholds in the detection of novel variants to understand what this parameter impacts.

Supplementary figures which schematically show how each method works would be very helpful to a general audience. Such figures are available on the swarm repository, thus this should be fairly straightforward to include.

== Results and Discussion ==

I would like to see some description of the impact of UCHIME and ChimeraSlayer on the datasets. How many reads were removed? Did the methods disagree? Were the hits removed confirmed to be chimeras? It would also be interesting to see what the impact of not performing a chimera check actually is on the OTU-clustering methods evaluated.

In general, it may be useful to abbreviate "sequence similarity networks" as this is mentioned very frequently.

The authors use the term "amplicon" for dereplicated sequences derived from amplicons here and in the methods. I would ask them differentiate the two.


162-172: It's difficult to support the usage of terms like "effective" - can we really claim efficiency when we're not sure about the accuracy? Stating these results with more neutrality would be best. Also, 'jumpling' to the conclusion that any method identifies more or less novel diversity is problematic - they may separate unique amplicon sequences based on their being classified or unclassified, but this doesn't mean they have found any meaningful diversity. Again, being neutral on this point or clearly explaining and demonstrating why this can be interpreted as real, novel diversity is necessary.

175: "..one third..." relative to the SSNs method I assume?

179-180: Using language like "split from the OTU" (and "over-splitting") is problematic - were they ever attached? More precise language would be good here. Also, verify that "97% divergent" is correct.

197-198: This computational limit should be described with more detail (e.g. memory requirments, CPU hours, etc).

218-225: This is why the "novel" adjective used throughout this manuscript needs to be qualified to reflect that we're dealing with an "initial classification of novel diversity".

240-242: I would be more cautious in claiming more accurate novel diversity discovery without actual ecological evidence. I would stop at stating the fact that the method did not fail to associate classified and unclassified OTUs which where more than 97% similar.

242-245: Perhaps I misunderstand this, but if a pairwise similarity matrix has been constructed, would accurate edges not be more of a certainty than an assumption?

252-254: rephrase for clarity, not sure that "common denominator" works here.

255-257: It would be very interesting to have an estimate of the total computational costs here and how they compare to the SSN approach.

275-276: Rephrase "With Swarm and OTUs containing both..." do the authors mean swarms and OTUs? or Swarm and other methods?

282-294: At this stage, I'm wondering why shortest path analysis is used rather than other network analysis techniques (as is noted in the next paragraph). The authors should provide rationale on this. Also, the focus of the manuscript has changed from evaluating 3 methods to considering the implications of graph theory on 2. This weakens the overall contribution, it would be much more interesting if the authors focused on the SSN and Swarm methods and more thoroughly evaluate network analyses. The issue with USEARCH and other centroid-based methods have been discussed in previous publications.

291.292: Do the authors mean "habitat" or "environment"? It sounds like the latter.

298-304: The conclusion confirms what I stated above: the main value here is in the analysis of the graphs generated by Swarm and SSN and this is only touched upon. This component needs to be strengthened for this publication to fulfill its (high) potential.

---

## Round 0.2 · accepted · Accept

With the changes made I now feel that your paper is acceptable for publication in PeerJ.

·

Basic reporting

No Comments

Experimental design

No Comments

Validity of the findings

No Comments

Additional comments

The authors have addressed all the concerns from the original submission and I think the paper is now ready for publication.